# Opposing Roles of FACT for Euchromatin and Heterochromatin in Yeast

**DOI:** 10.3390/biom13020377

**Published:** 2023-02-16

**Authors:** Shinya Takahata, Yota Murakami

**Affiliations:** Department of Chemistry, Faculty of Science, Hokkaido University, Sapporo 060-0810, Japan

**Keywords:** FACT, Spt16, Pob3, Nhp6, SBF, heterochromatin, Swi6, Clr4, H3K9me, cell cycle

## Abstract

DNA is stored in the nucleus of a cell in a folded state; however, only the necessary genetic information is extracted from the required group of genes. The key to extracting genetic information is chromatin ambivalence. Depending on the chromosomal region, chromatin is characterized into low-density “euchromatin” and high-density “heterochromatin”, with various factors being involved in its regulation. Here, we focus on chromatin regulation and gene expression by the yeast FACT complex, which functions in both euchromatin and heterochromatin. FACT is known as a histone H2A/H2B chaperone and was initially reported as an elongation factor associated with RNA polymerase II. In budding yeast, FACT activates promoter chromatin by interacting with the transcriptional activators SBF/MBF via the regulation of G1/S cell cycle genes. In fission yeast, FACT plays an important role in the formation of higher-order chromatin structures and transcriptional repression by binding to Swi6, an HP1 family protein, at heterochromatin. This FACT property, which refers to the alternate chromatin-regulation depending on the binding partner, is an interesting phenomenon. Further analysis of nucleosome regulation within heterochromatin is expected in future studies.

## 1. Introduction

Gene expression requires the binding of transcriptional activators that recognize and bind specific upstream activating sequences of DNA on the promoter of each gene or enhancer of distal sites [1,2,3]. In most cases, chromatin remodeling factors, histone chaperones, and histone acetyl-transferase complexes are recruited by gene-specific transcriptional activators to loosen the core promoter chromatin around the TATA-box and the transcriptional start site for the formation of a preinitiation complex with the general transcription factors, TFIIA, TFIIB, TFIID, TFIIE, TFIIF, TFIIH, and RNA polymerase II, adjacent to the +1 nucleosome [4,5,6,7,8]. The sequential scheme of this transcription initiation indicates the repressive feature of the chromatin structure itself against gene expression, and that the regulatory mechanism of relaxing and closing the chromatin structure is closely related to the regulation of gene expression [9,10]. The landscape of nucleosome occupancy is analyzed by ChIP-seq or MNase-seq, continuously revealing the genome-wide positioning of nucleosomes and profiling of gene expression [11,12,13,14]. In terms of chromatin structure, there is euchromatin with a loose chromatin structure and heterochromatin with a complex higher-order chromatin structure [15], with post-translational modifications of histone proteins playing a key role in maintaining each structure [16,17]. It has been reported that the SIR complex, Sir2, Sir3, and Sir4, is responsible for the heterochromatin formation of the MAT locus, subtelomeres, and rDNA regions depend on a DNA element called “silencer” in budding yeast [18,19]. The heterochromatin formed in this way is somewhat unique and differs from the heterochromatin in other eukaryotes, which is epigenetically formed and maintained by the post-translational modifications of histones. SIR complex-mediated silencing is assumed to be more stably maintained by DNA-binding factors. In addition, there have been no reports that FACT contributes to this SIR complex-dependent silencing of budding yeast heterochromatin so far. In fission yeast, heterochromatin is formed and maintained through high-histone H3K9 methylation and low-histone H3K4 methylation by histone H3K9 methyltransferase, histone H3K4 de-methylase, and HP1, as in mammal cells [20,21,22,23,24,25,26,27]. A histone H3K9 methyltransferase and HP1 family proteins are conserved as in other eukaryotes, and are responsible for the constitutive heterochromatin at the MAT locus, pericentromeres, and subtelomeres [21]. In addition, recent studies reported that H3K9me islands are scattered on chromosome arms [28,29]. The FACT complex has been shown to stabilize the constitutive heterochromatin by working in concert with HP1/Swi6 in fission yeast [30,31,32]. While the chromatin remodeling models of FACT for transcriptional stimulation on euchromatin have been proposed to date, a completely new mechanism of chromatin silencing by FACT that represses nucleosome dynamics on heterochromatin will become a subject of discussion.

## 2. FACT Plays Multifunctional Roles in Transcriptional Regulation

The FACT complex, a heterodimer of Spt16 and SSRP1, was isolated as an RNA polymerase II transcriptional elongation factor required for efficient chromatin transcription [33,34,35,36]. Unlike other chromatin remodeling factors, FACT has no ATPase domain and performs chromatin remodeling in an ATP-independent manner. Recent studies have reported that nucleosome regulation by interplay between FACT and Chd1, an ATP-dependent chromatin remodeling factor, plays an important role in RNA polymerase II transcriptional initiation at +1 nucleosome [37,38]. In addition to RNA polymerase II-mediated transcription, TFIIIC/TFIIIB may be involved in a physical interaction with the FACT complex for the RNA polymerase III transcribed tRNA genes in budding yeast [39]. The nucleosome-binding activity of human FACT is low in vitro, but through some destabilization in the contact between core histones and nucleosomal DNA, FACT starts interacting with the core region of the histone covered by DNA [40,41]. FACT is a highly conserved histone chaperone between divergent eukaryotic species (Figure 1).

In yeast, it consists of an Spt16/Pob3 heterodimer and the high-mobility group box (HMGB) protein Nhp6 [42,43,44,45]. Spt16 was originally isolated as *CDC68*, a gene responsible for causing G1 arrest [46]. Pob3 and Nhp6 are bipartite analogs of SSRP1 [47,48]. The intracellular roles of FACT in budding and fission yeast appear to be somewhat different. First, the copy numbers of the budding yeast *NHP6* gene and the fission yeast *nhp6+* gene are different: there are two copies of *NHP6* in budding yeast, *NHP6A* and *NHP6B* [49], whereas there is a single copy of *nhp6+* in fission yeast [50]. Second, *SPT16* and *POB3* are essential genes, whereas *NHP6A/B* are nonessential genes in budding yeast [44,51]. In fission yeast, *spt16+* is an essential gene, whereas *pob3+* and *nhp6+* are nonessential genes [50], allowing the disruption of *pob3+* and *nhp6+*, which are functional bipartite analogues of SSRP1 in multicellular organisms, exceptionally through eukaryotic species. Considering that the function of Spt16 is required even in the absence of Pob3 and Nhp6, it is likely that it exerts a certain chromatin regulatory role, which is specific to Spt16 in fission yeast. In suggesting a role for each FACT component, human SSRP1 has Spt16-dependent and independent roles in transcriptional regulation [52,53]. In budding yeast, Pob3 forms a stable heterodimer with Spt16 via their dimerization domains [43,48]. Biochemical studies have exhibited that Nhp6 plays an essential role in binding the Spt16/Pob3 heterodimer to the nucleosome [45], with the required amount of Nhp6 appearing to be stoichiometrically in excess to that of the Spt16/Pob3 heterodimer [54]. This might suggest that when the HMGB DNA-binding domain is fused to the FACT, as in SSRP1, it enhances nucleosome recognition for the efficient H2A/H2B dimer eviction from the nucleosome. Another FACT isoform, in which Pob3 and Nhp6 are expressed as a fusion SSRP1 protein, has also been analyzed in vivo. In budding yeast, Nhp6 is expressed from *NHP6A* and *NHP6B*, and strains in which both *NHP6A* and *NHP6B* are simultaneously disrupted showed growth defects. Under this condition, the expression of the *POB3-NHP6* fusion gene was found to complement the growth defect shown by the *nhp6ab∆* strain [47]. Moreover, the Pob3-Nhp6-fused FACT has been reported to be involved in nucleosome regulation, as indicated by biochemical analyses. Single or multiple HMGB modules were fused to Pob3 to mimic SSRP1 for evaluating its nucleosome-binding capacity. Human SSRP1 and a yeast Pob3-Nhp6 fusion both required free Nhp6 to support nucleosome reorganization. This result indicated that a single intrinsic DNA-binding HMGB was not sufficient for intact FACT nucleosome reorganizing activity, whereas triple HMGB modules at the *C*-terminus of Pob3 supported FACT activity without free Nhp6. However, this FACT variant was not efficiently released from nucleosomes, in turn exhibiting toxicity in yeast [55]. Recent cryo-EM structure analysis revealed that human Spt16 bound to histones in a subnucleosome and tethered H2A/H2B through its C-terminal acidic tail by acting as a placeholder for DNA, with no electron density being observed at the HMGB domain of SSRP1 [56]. Phosphorylation of the Spt16 C-terminal acidic tail is required for its binding to H2A/H2B in the nucleosome [57,58], suggesting the involvement of CKII [59,60]. FACT was also reported to displace H2A/H2B dimers from the nucleosome through the tandem PH domain of Spt16 and histone H3/H4-binding of the Spt16 peptidase-like domain with the help of Nhp6 [61,62,63]. Apart from its DNA-binding activity, cryo-EM analysis revealed that Nhp6 binds to both C-terminal acidic tails of Spt16 and Pob3 to unfold the FACT complex structure for the activation of efficient nucleosome reorganization [64]. These results suggested the importance of Nhp6 for chromatin remodeling [56].

## 3. Working Models of FACT for Nucleosome Dynamics in Fission Yeast

Various molecular models have been proposed to explain the means by which FACT transforms the nucleosome [57,64,65], but few molecular models have been proposed for how FACT regulates chromatin silencing. In the case of the fission yeast FACT, the histone H3/H4-binding activity of the peptidase-like domain at the N-terminus of Spt16, the histone H2A/H2B chaperone activity of the tandem PH domain in the central region of Spt16, the histone H3/H4-binding activity of the tandem PH domain of Pob3, and the DNA-binding activity of Nhp6 are thought to play key roles in nucleosome recognition [30,61,62,66]. Accordingly, two different models by which fission yeast FACT binds to the mononucleosome or dinucleosome is shown in Figure 2. In the case of binding to the mononucleosome, the peptidase-like domain of Spt16 and the tandem PH domain of Pob3 bind to the two histone H3/H4 dimers present in the mononucleosome via their dimerization domains, respectively (Figure 2A).

After binding of the Spt16/Pob3 heterodimer to the mononucleosome, the acidic tail at the C-terminus of the two proteins competes with the nucleosomal DNA on the surface of histone H2A/H2B. Following this competition, Nhp6 binds to the fluctuated DNA and promotes the divergence of histone H2A/H2B from DNA in the nucleosome, with the tandem PH domain of Spt16 depositing the histone H2A/H2B dimer from the octasome and transforming it to a hexasome or tetrasome [67]. Studies have already reported the histone chaperone activity of both human and yeast FACT for histone H2A/H2B [68], suggesting the induction of a transient dynamic change in chromatin regulation by a similar process of nucleosome conformational change. Meanwhile, the histone-binding properties of the peptidase-like domain of Spt16 and the tandem PH domain of Pob3 have suggested their binding to the dinucleosome (Figure 2B). In this case, the peptidase-like domain of Spt16 and the tandem PH domain of Pob3 are expected to act separately on two neighboring nucleosomes to bridge them, such as HP1; however, the mechanism by which the tandem PH domain of Spt16 acts on histone H2A/H2B in the nucleosome remains undetermined. As we currently lack any biochemical or structural data on the mechanism of action of FACT for the dinucleosome, therefore the histone H2A/H2B chaperone activity of FACT should be determined in the dinucleosome regulatory case.

## 4. SBF Recruits FACT for Promoter Chromatin Activation in Budding Yeast

Previous studies have revealed that FACT dynamically alters the chromatin structure, transiently evicting nucleosomes for the passage of RNA polymerase II in vivo [63]. Following nucleosome eviction and the passage of RNA polymerase II, FACT deposits nucleosomes to close the transiently loosened chromatin structure [69,70]. In addition to functioning as a transcriptional elongation factor, previous studies have demonstrated that FACT binds to the G1/S START transcription factors SBF and MBF (Figure 3A), which are analogs of mammal E2F in budding yeast. SBF enters the nucleus in late M/early G1 phase [71,72,73] and binds to the G1 gene promoters. In turn, the SBF-recruited FACT transiently evicts nucleosomes from the promoter of G1/S regulatory genes before initiation of transcription by RNA polymerase II [74,75,76]. In budding yeast, SBF acts at the START checkpoint in the G1/S phase, regulating the expression of the *CLN1* and *CLN2* genes [77,78,79] (Figure 3A). Cell cycle-related gene transcription is regulated by the competition between positive and negative regulators of chromatin. In early G1 phase, the activity of SBF is repressed by Whi5 after binding to the “CACGAAAA” promoter element in the UAS until it is activated to initiate transcription at the proper timing. Whi5 recruits the histone deacetyl-transferase complex Rpd3(L) to keep the promoter chromatin in a silent state [74]. Cyclin kinase Cdk1/Cln1,2 phosphorylates Whi5 during the progression of the G1 phase to remove it from SBF [80], which is then converted to its activated state [81,82]. After the removal of Whi5 and Rpd3(L) from the promoter, the SBF-recruited FACT functions to change the promoter chromatin structure [74]; however, the detailed molecular mechanism by which FACT recognizes SBF/MBF remains undetermined. The expression timing of G1 genes during G1 phase is also different, with variations observed during the transition from early G1 to late G1/S. Even though they are regulated by the same transcription factor, SBF/MBF, it is assumed that the reason for this is the differences in the chromatin structure of the promoters of each gene. In addition to the G1 cyclin gene, the chromatin structure of the homothallic switching (*HO*) gene promoter is regulated by SFB and FACT (Figure 3B). The *HO* gene on chromosome IV encodes the Ho endonuclease.

Budding yeast strains commonly used in laboratories contain a mutation that results in defective nuclease activity in vitro and in vivo [83]. Wildtype Ho endonuclease induces a double-strand break that targets a DNA element in the MAT locus, causing mating-type switching via gene conversion of the MAT decision cassette [84,85,86,87]. This phenomenon occurs asymmetrically during cell division, with expression of the *HO* gene in the mother cell and transcriptional repression of the *HO* gene in the daughter cell. This asymmetry is generated by Ash1, a component of Rpd3(L) HDAC [88,89] that is mainly expressed in daughter cells, and binds to the promoter of the *HO* gene, thereby strongly repressing transcription [76,90,91,92,93,94]. In addition to this asymmetric expression, the expression of *HO* needs to be strictly regulated in the mother cell. The promoter structure of the *HO* gene is relatively long and complex compared with that of common yeast genes, and consists of two sequential regions, URS1 and URS2 (Figure 3B). The combination of URS1 and URS2, approximately 1.0 kbp each, regulates cell cycle-dependent transcriptional initiation [75,95,96,97,98]. URS1 contains two binding sites for Swi5, which is expressed at the M/G1 phase boundary. Swi5 is phosphorylated by the Cdk1 kinase and is transported into the nucleus from the end of M to the early G1 phase [99,100,101]. An SBF-binding site has also been identified at the 3′ side of URS1; however, mutation of this SBF-binding sequence does not affect the expression of the *HO* gene, suggesting that it is not a functionally essential element [76]. The SAGA complex, Swi/Snf complex, and SRB mediator complex are then recruited onto URS1 by Swi5 to loosen the chromatin structure of URS1. In turn, SBF and FACT loosen the chromatin structure from the 3′ side of URS1 to the 5′ side of URS2 and recruit additional SBF activators for the recruitment of the SAGA complex, Swi/Snf complex, and SRB mediator complex downstream of URS2 and TATA-box [74,75,95,96]. FACT is assumed to be the factor that causes the nucleosome eviction from the 3′ side of URS1 to the 5′ side of URS2 in the sequence of chromatin conformational changes involved in this transcriptional activation.

## 5. Wave of Nucleosome Eviction, and the Site Where FACT Functions in *HO* Promoter in Budding Yeast

A schematic representation of the FACT working region in the *HO* promoter and the dynamic changes in chromatin along cell cycle progression is shown in Figure 4. In wildtype budding yeast strains, nucleosome eviction at URS1 is triggered by the Swi5 activator, and the recruited SAGA complex, Swi/Snf complex, and SRB mediators. FACT-induced nucleosome eviction is then triggered from downstream URS1 to upstream URS2, with the wave of nucleosome eviction being propagated downstream to URS2 and the core promoter, where the chromatin around the TATA-box is finally opened to promote the transcription of the *HO* gene by RNA polymerase II (Figure 4A). The nucleosomes of URS1 are quickly repositioned, presumably due to polyubiquitination and proteolysis of Swi5 [100]. However, in the FACT mutant yeast strain, nucleosomes are evicted from URS1 by the Swi5 activator and the recruited SAGA complex, Swi/Snf complex, and SRB mediators, as in wildtype, but this eviction is not propagated downstream from the URS2 to the TATA-box during cell cycle progression (Figure 4B). ChIP analysis of FACT exhibited a biased binding pattern to the upstream of URS2 of the *HO* gene promoter, suggesting that FACT does not bind to the promoter solely through the SBF recruitment [75]. The reason for this biased promoter-binding activity of FACT remains unclear, and there might be a characteristic chromatin structure upstream of URS2 that FACT prefers.

## 6. FACT-Dependent Heterochromatic Silencing in Fission Yeast

Similar to many other eukaryotes, heterochromatin in fission yeast is formed in a histone H3K9 methylation-dependent manner. The mechanism of formation of constitutive heterochromatin at centromeres, subtelomeres, and the MAT locus is very complex [21,102,103,104], with the molecular mechanism of heterochromatin formation being distinct in these three regions [105]. Histone H3K9 methylation-dependent higher-order chromatin structures cannot be stably maintained unless the various effector factors function at the correct timing. Methylation of histone H3K9 is the most important factor for heterochromatin formation. Although multicellular eukaryotes possess multiple histone H3K9 methyltransferases, in fission yeast, Clr4 is the sole source of methylase of histone H3K9 via its SET domain [106]. A recent study revealed that automethylation of Clr4 stimulates its enzymatic activity and maintains its epigenetic stability [107]. At least two recruitment mechanisms are known for Clr4 in the establishment of pericentromeric heterochromatin. One is the direct association of Clr4 with HP1/Swi6 [108], and the other is an RNAi-dependent recruitment onto heterochromatin [109,110] (Figure 5). Although heterochromatin formation and transcription of noncoding RNAs (ncRNAs) sound contradictory, HP1/Swi6 is strongly bound by Epe1, a JmjC protein [111,112]. Epe1 carries the acidic activation domain at the N-terminus and stimulates the transcription of heterochromatic ncRNAs by RNA polymerase II [112]. This transcription in the heterochromatin is assumed to be slow and suspendable and creates a scaffold retaining the nascent ncRNA on heterochromatin for RNAi-related effectors on heterochromatin [113,114,115,116].

In addition to the SET domain, Clr4 itself also has a chromodomain (CD) at its *N*-terminus that recognizes histone H3K9me, and following recognition exerts its self-propagation ability to methylate H3K9 on the adjacent nucleosome [25,117]. Swi6 and Chp2 are known as fission yeast HP1, which bind to H3K9me-containing nucleosomes, forming homodimers via their chromo-shadow domain (CSD) [118,119]. Heterochromatin is stably maintained by two homodimers, Swi6 and Chp2 [120,121], which attract different silencing effectors [122,123,124], with Swi6 being a more versatile HP1, potentially important for the formation and maintenance of stable heterochromatin. In addition to the different roles of the two HP1 family proteins in fission yeast, post-translational modifications of HP1 also affect HP1 heterochromatin formation. For instance, Swi6 has been reported to be phosphorylated, and mutations at the phosphorylation site were reported to disrupt heterochromatin formation [125,126,127].

Genetic analysis and ChIP-qPCR showed that fission yeast strains lacking *pob3+* (*pob3∆*) had comparable levels of histone H3K9 methylation and Swi6 localization in the heterochromatic region to those of the wildtype strain, but with high levels of heterochromatic expression of ncRNAs. Phenotypic analysis of the *pob3∆* strain indicated that heterochromatic silencing was defective in heterochromatin without a significant loss of levels of histone H3K9 methylation and HP1/Swi6-binding [30,128]. ChIP analysis of Spt16 exhibited that the binding level of Spt16 to the heterochromatic region in the *pob3∆* strain was decreased to half that of the wildtype, suggesting the existence of a Pob3-independent recruitment mechanism of Spt16 onto heterochromatin. Genetic analysis also revealed that *pob3∆swi6∆* double disruption exhibited an additive silencing defect compared with that shown in each of the *pob3∆* and *swi6∆* single-mutant strains [30]. Therefore, we assumed that the recruitment of Spt16 onto heterochromatin is partially dependent on Swi6. To test this hypothesis, we performed biochemical analysis using recombinant Spt16 and the fission yeast HP1 family, Swi6 and Chp2. We found that the peptidase-like domain of Spt16 directly binds to the dimerized chromo-shadow domain (CSD) of Swi6, but not to Chp2-CSD [30,120,129]. Although the “PxVxL/I” hydrophobic amino acid sequence of the CSD-binding motif is necessary for stable CSD-binding [130,131,132], this motif is not conserved in the peptidase-like domain of Spt16. Further Spt16-Swi6-binding experiments revealed that the binding of Spt16 and Swi6 was easily compromised by increasing the salt concentration in the binding buffer in vitro, suggesting the existence of a novel binding mode between Spt16 and Swi6.

To elucidate this binding mode between Spt16 and Swi6, we carefully compared the primary sequences of Swi6 and Chp2 in the CSD and found a difference in the β1–β2 connecting loop, which forms the protrusion in the CSD homodimer [30]. We identified a charge-biased “RKDD” cluster in the β1–β2 connecting loop of Swi6-CSD, but no such charge-biased cluster in the β1–β2 connecting loop of Chp2-CSD. Other HP1 family proteins were also examined for the presence of charge-biased clusters in the β1–β2 connecting loop, but no charge-biased clusters were found. A physical interaction between HP1c and SSRP1 for transcriptional activation on euchromatin has been reported in *Drosophila melanogaster* [133], but physical interactions between HP1 and Spt16 have not been reported in other species so far. The charge-biased cluster in the β1–β2 connecting loop of the Swi6-CSD might be a specific property of the formation of heterochromatin in fission yeast. These data have suggested that the “RKDD” sequence in the β1–β2 connecting loop of Swi6-CSD might function as a binding surface of the peptidase-like domain of Spt16. To this end, a recombinant Swi6-4A mutant (“RKDD” to “AAAA”) was used to test the binding activity of Spt16 in vitro. As expected, Swi6-4A lost its ability to bind to the peptidase-like domain of Spt16 [30]. Interestingly, heterochromatin was significantly disordered in Swi6-4A mutant fission yeast. Although FACT targets the β1–β2 connecting loop of Swi6, other effectors might bind to the “RKDD” sequence in the β1–β2 connecting loop of Swi6-CSD in fission yeast. This scenario requires further examination in future heterochromatin studies. To evaluate the effect of eliminating FACT in heterochromatin, a peptidase-like domain-truncated Spt16 was expressed in a *pob3∆* strain. Different from the *pob3∆* strain, a major decrease in the levels of histone H3K9 methylation and Swi6-binding was observed [30]. This indicated that FACT plays a critical role in the establishment and maintenance of heterochromatin.

## 7. Mechanism of Action of FACT on Nucleosomes within Heterochromatin for Formation and Maintenance of Heterochromatin in Fission Yeast

Our analysis revealed the molecular mechanism by which FACT is recruited onto heterochromatin for the dynamic regulation of the H2A/H2B dimer and optimal management of nucleosomes in heterochromatin [30]. The H2A.Z/H2B dimer contributes to a certain extent to this heterochromatic silencing [134]. Other groups have reported that FACT strongly suppresses histone turnover [31,32,135], and the mechanism by which FACT regulates nucleosomes in heterochromatin is an important aspect to be considered. To predict the means by which FACT regulates heterochromatic nucleosomes in fission yeast, we proposed a hypothetical model, as shown in Figure 6.

Within the heterochromatin, two nucleosomes are bridged by the HP1/Swi6 heterodimer via histone H3K9me (Figure 6A). Structural analysis revealed that it is not sufficient to enumerate the dinucleosome with a HP1 homodimer alone, and the linker DNA is exposed to allow the access of silencing effectors [136]. In fission yeast, the Swi6-CSD homodimer physically binds to the peptidase-like domain of Spt16 to recruit FACT onto the heterochromatin [30]. Concomitantly, the tandem PH domain of Pob3 binds to the H3/H4 dimer in a heterochromatic nucleosome (Figure 6B). After the recruitment of FACT by Swi6, the peptidase-like domain of Spt16, which recognized Swi6-CSD, shifts its scaffold to histone H3/H4 in the nucleosome [30]. This scaffold shift can be divided into two modes: binding to the dinucleosome as a bridge (Figure 6C) and binding to the mononucleosome (Figure 6D), shown as euchromatic nucleosome regulation in Figure 2. In the heterochromatic dinucleosome bridging model, FACT helps Swi6 enhance nucleosome condensation and the formation of a higher-order chromatin structure (Figure 6C). In the heterochromatic mononucleosome-binding model, the peptidase-like domain of Spt16 and the tandem PH domain of Pob3 clip the histone H3/H4 tetramer in the heterochromatic nucleosome to stably tether FACT on a single nucleosome (Figure 6D). Under the stable FACT-nucleosome binding, deposition and reposition of the histone H2A/H2B dimer occur via the chaperon activity of the Spt16 tandem PH domain [61]. Histone H2A/H2B ChIP analysis in the *pob3∆* strain indicated that reposition of the H2A/H2B dimer was dependent on the recognition of the stable H3/H4 dimer in nucleosomes by the tandem PH domain of Pob3 [30].

## 8. Summary and Perspective

FACT is an important transcription stimulating factor that transiently relaxes the chromatin structure through its histone H2A/H2B chaperone activity. The transcription-activating properties of FACT are especially demonstrated in START, the checkpoint of the G1/S phase, through binding to SBF/MBF in budding yeast [74,75,76]. However, the mechanism by which FACT contributes to START activation by the mammalian E2F family remains unclear. In addition to transcriptional activation in budding yeast, studies using fission yeast have shown that FACT contributes to the formation and maintenance of histone H3K9-mediated heterochromatin [30]. In particular, the molecular mechanism by which FACT is recruited onto heterochromatin was previously analyzed in detail by the authors, and the means by which FACT regulates nucleosomes in heterochromatin will be a subject of future research.

## Figures and Tables

**Figure 1 biomolecules-13-00377-f001:**
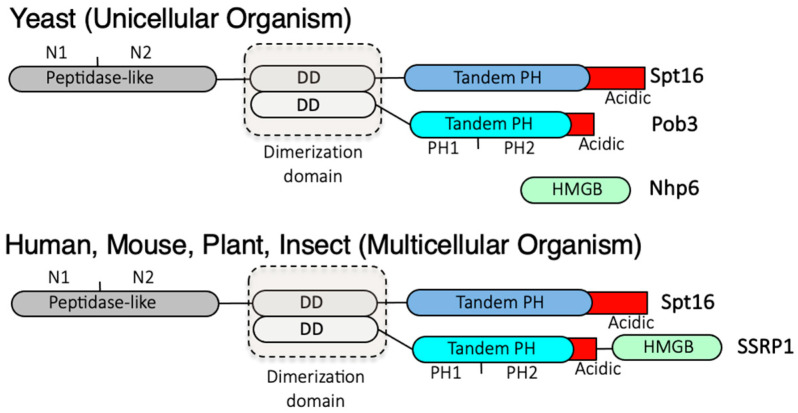
Schematic structures of FACT complex in yeast and higher eukaryotes. N1: subdomain of peptidase-like domain 1, N2: subdomain of peptidase-like domain 2, DD: dimerization domain, Tandem PH: tandem pleckstrin homology domain, PH1: pleckstrin homology domain 1, PH2: pleckstrin homology domain 2, Acidic: acidic amino acid cluster, HMGB: high-mobility group box.

**Figure 2 biomolecules-13-00377-f002:**
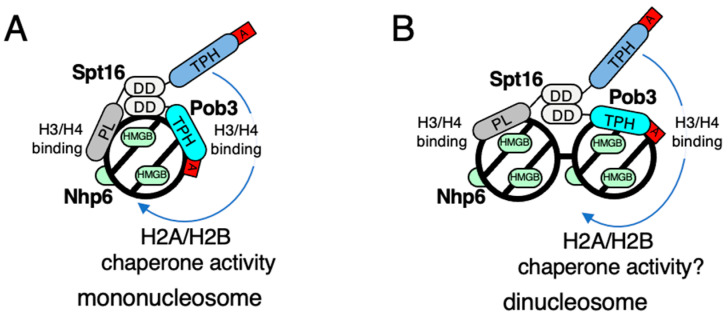
FACT working model for the mononucleosome and dinucleosome. PL: peptidase-like domain, TPH: tandem PH domain, DD: dimerization domain, A: acidic cluster. (**A**) Yeast FACT on mononucleosome. FACT functions as a histone chaperone in this scenario. (**B**) Yeast FACT on dinucleosome. Whether FACT functions as a histone chaperone in this scenario remains unknown.

**Figure 3 biomolecules-13-00377-f003:**
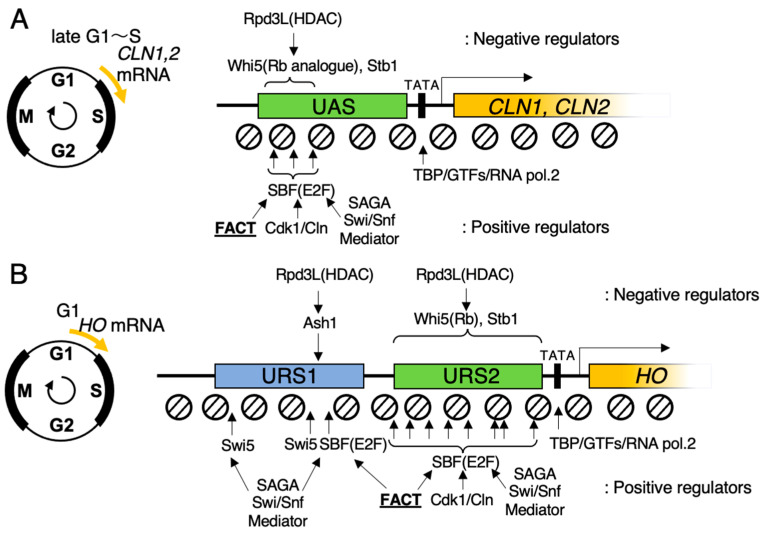
Schematic diagrams of SBF-regulated gene promoter. UAS: upstream activating sequence, URS1: upstream regulatory sequence 1, URS2: upstream regulatory sequence 2. (**A**) Promoter structure of *CLN1* and *CLN2* genes. Three SBF-binding sites are annotated as arrows. Positive and negative regulators of gene expression are shown below and above the schematic of the gene structure, respectively. FACT is assigned an underline. (**B**) Promoter structure of the *HO* gene. Two Swi5-binding sites, one SBF-binding site, and one Ash1-binding site are annotated in URS1. Eight SBF-binding sites are annotated in URS2. Positive and negative regulators of gene expression are shown below and above the schematic of the gene structure, respectively. FACT is assigned an underline.

**Figure 4 biomolecules-13-00377-f004:**
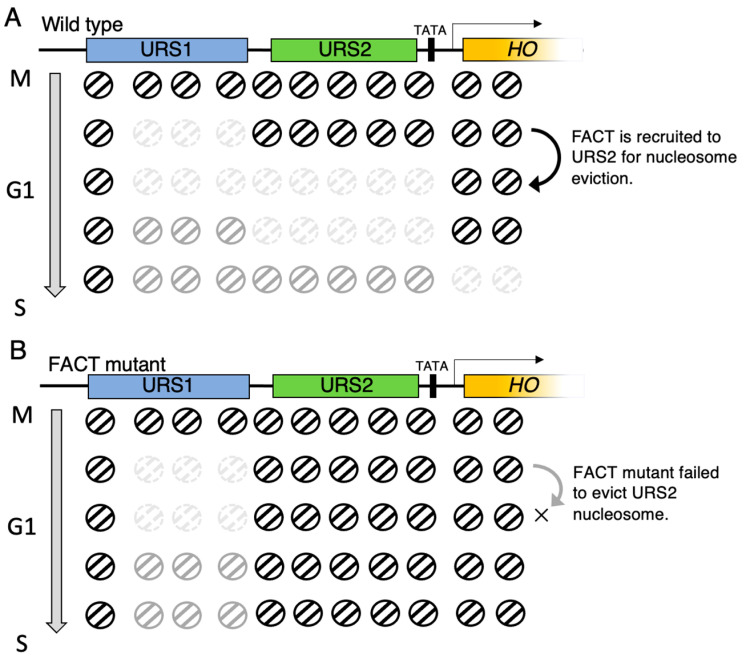
Wave of nucleosome deposition and reposition at the *HO* promoter during cell cycle progression. Based on *GALp::CDC20* arrest/release experiment, nucleosome dynamics were analyzed by time-course histone H3 ChIP-qPCR during M to S phase transition. URS1: upstream regulatory sequence 1, URS2: upstream regulatory sequence 2. (**A**) Schematic representation of nucleosome dynamics along the *HO* promoter in wildtype cells. (**B**) Schematic representation of nucleosome dynamics along the *HO* promoter in FACT mutant cells.

**Figure 5 biomolecules-13-00377-f005:**
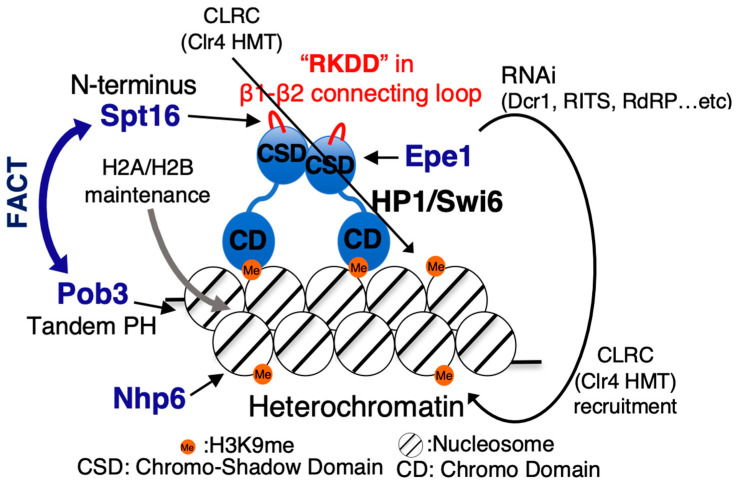
Schematic diagrams of the FACT recruitment model at pericentromeric heterochromatin.

**Figure 6 biomolecules-13-00377-f006:**
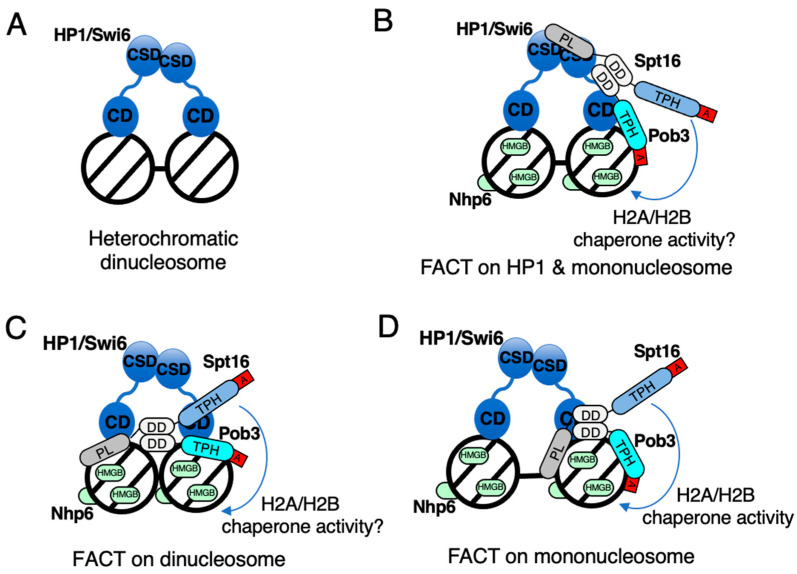
Hypothetical FACT working model on heterochromatic nucleosomes. PL: peptidase-like domain, TPH: tandem PH domain, DD: dimerization domain, A: acidic cluster, CD: chromodomain, CSD: chromo-shadow domain. (**A**) HP1/Swi6 bridge dinucleosome. (**B**) FACT is recruited by a Swi6-CSD homodimer via the peptidase-like domain of Spt16. (**C**) HP1/Swi6 and FACT cooperatively bridge dinucleosome. (**D**) FACT binds to heterochromatic mononucleosome.

## Data Availability

Not applicable.

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
