# Peer review of "Opposing Roles of FACT for Euchromatin and Heterochromatin in Yeast"

_biomolecules, 2023, doi:10.3390/biom13020377_

Round 1

Reviewer 1 Report

This review article focusses on the opposite roles of the FACT complex in euchromatin and heterochromatin in yeast. It addresses one of the most relevant questions of the field by discussing how different interacting partners of a protein complex can display specialized functions. 

The authors embrace the topic comparing FACT performance in budding and fission yeast. They discuss the contrasting roles displayed by the FACT complex when working in euchromatin and heterochromatin in yeast based on previous literature alluding to relevant current articles on the topic. They make use of the extensive previous knowledge about FACT involvement on gene activation in budding yeast and its achievements in chromatin compaction in fission yeast. Furthermore, they present models that illustrate how FACT display different activities when binding either to mono nucleosomes or nucleosomes in fission yeast. Overall, the manuscript is substantial and reads well. However, I have a few comments intending to improve the specific sections that I expand below.

Specific points

The introduction is a bit too short. I would like the authors to expand a little more about characteristics of euchromatin and heterochromatin in budding and fission yeast. 

Some sections only focused on one or the other biological model. It would be good to have the manuscript structured in a way that can explain why some topics are only discuss for one or the other system to make it easier to read to a broader audience. Additionally, it would be nice to compare both yeast models whenever it is possible as explained for section 3.

Section 3 “Working models of FACT for nucleosome dynamics”. This section addresses a very relevant and point,  but is poorly written.

 First, this section is very poor in references. They start by referring to a paper that focus on human model, moving to discuss FACT activities on fission yeast without calling any specific reference. The authors should support better their claims on the compared models, since proper references for fission yeast models are missing in this section. My second concern is that they only discussed FACT working models for nucleosome dynamics in fission yeast, however there are several articles discussing FACT involvement on this matter in budding yeast as well. It would be interesting to show some comparison or discuss a few facts of the current knowledge about FACT role on nucleosome dynamics in budding yeast. A few suggestions to give some, are to include PMID: 30681413, PMID: 35981082, PMID: 33277439. I think these additions will improve the manuscript. 

When talking about nucleosome dynamics is hard to avoid thinking about ATP-dependant chromatin remodelling. I think some additional paragraphs discussing the interplay between FACT and Chd1 PMID: 33846633PMID: 33590780, PMID: 34380014 can also add to it.

Seccion 6. In the last paragraph some references are missing

Author Response

Comments from reviewer #1:

The introduction is a bit too short. I would like the authors to expand a little more about characteristics of euchromatin and heterochromatin in budding and fission yeast. 

Authors sincerely thank the reviewers for careful reading and comments that improved this manuscript. Authors expanded introduction especially focusing on the relationship between heterochromatin and FACT in budding yeast and fission yeast.

Some sections only focused on one or the other biological model. It would be good to have the manuscript structured in a way that can explain why some topics are only discuss for one or the other system to make it easier to read to a broader audience. Additionally, it would be nice to compare both yeast models whenever it is possible as explained for section 3.

Authors understand the reviewer’s point to compare the nucleosome regulation models of budding yeast and fission yeast especially in terms of chromatin silencing, but several important information is still missing in budding yeast FACT research. Fission yeast Spt16 and Pob3 were well characterized by genetic, biochemical, and structural analysis. But budding yeast peptidase-like domain of Spt16 and the C-terminal acidic domain of Pob3 are not well characterized by biochemical and structural analysis. In addition, human peptidase-like domain of Spt16 and the C-terminal domain of SSRP1 were disordered in cryo-EM analysis. Therefore, authors made working models based on fission yeast FACT in this manuscript.

Section 3 “Working models of FACT for nucleosome dynamics”. This section addresses a very relevant and point, but is poorly written.

 First, this section is very poor in references. They start by referring to a paper that focus on human model, moving to discuss FACT activities on fission yeast without calling any specific reference. The authors should support better their claims on the compared models, since proper references for fission yeast models are missing in this section.

Authors added additional references in section3. In this section, we are not arguing for nucleosome organization by FACT, which has been already proposed as a model, but for a high degree of chromatin-repressive activity by FACT found in our fission yeast experiments. To avoid misinterpretation by the reader, we have added the sentence "but few molecular models proposed for how FACT regulates chromatin silencing” in section 3.

My second concern is that they only discussed FACT working models for nucleosome dynamics in fission yeast, however there are several articles discussing FACT involvement on this matter in budding yeast as well. It would be interesting to show some comparison or discuss a few facts of the current knowledge about FACT role on nucleosome dynamics in budding yeast. A few suggestions to give some, are to include PMID: 30681413, PMID: 35981082, PMID: 33277439. I think these additions will improve the manuscript. 

PMID: 30681413, PMID: 35981082 and PMID: 33277439 were included for additional references.

When talking about nucleosome dynamics is hard to avoid thinking about ATP-dependant chromatin remodelling. I think some additional paragraphs discussing the interplay between FACT and Chd1 PMID: 33846633, PMID: 33590780, PMID: 34380014 can also add to it.

 Importance of interplay between FACT and Chd1 are now included in section 2.

Seccion 6. In the last paragraph some references are missing

References are added in the last paragraph of Section 6.

Reviewer 2 Report

The authors present a wide-ranging description of the FACT complex, with a focus in yeast model organisms. They begin by discussing general roles of chromatin compaction dynamics in heterochromatin and euchromatin, and the importance in transcriptional control across eukaryotes. Their focus narrows to consider the FACT complex, its composition and roles. The authors describe primarily work from budding yeast and fission yeast as counterpoints and tools to understand FACT’s roles in transcriptional regulation. Despite the differences from metazoan models, the authors nicely demonstrate how the complimentary yeast works point to FACT functions in multicellular organisms. FACT binding to nucleosomes is approached in Figure 2, and then recapitulated Figure 6 considering heterochromatic nucleosomes. Two separate transcriptional modes of FACT function (SBF, HO promoters) are detailed to discuss similarities and complexities, particularly as related to E2F-dependent human transcription at START.  The authors describe not only the conclusions from published work (theirs and other labs), but they also indicate key areas where details are unknown and future lines of inquiry. References are well used, although page 4 lines 122-141 feel under-referenced and missing some contextual markers. This work will add summary and future considerations for readers.

Comments:

1. The manuscript would benefit from some additional editing to correct typographical errors. For example, p9 line 319 “... certain extend [extent]”, and several long sentences that could be edited for clarity. Generally, the manuscript reads clearly.

2. There is a great deal of information dealing with Figure 1 and associated text in Section 2. Section 2 becomes rather dense on page 3 when describing human and budding yeast similarities. Might the authors consider a table to summarize the functions of subunits between yeasts and metazoans?  It may help clarify overall statements and alignment of roles/similarities/differences described.

3. Figure 2- might the authors consider adding labels for subunits of FACT components/domains or an associated legend to reinforce connections of subunits with nucleosomes. This will help a broader readership to understand the structures described, particularly for HMGB-Nhp6.

4. Figure 6 (and dinucleosome of figure 2)- linker DNA is mentioned; the authors might include indication of linker DNA in the diagram, along with labeled subunits.

5. Additional summary of the applications of the work into humans/multicellular organisms may be considered in the summary and perspective (Section 8). Since FACT has both heterochromatic and transcriptional-activating roles (and across 2 yeast models), a few additional sentences to link these to human work (particularly given the differences from yeast to human, but also impact for human biomedical applications) could be particularly valuable.

Author Response

Point by point response for reviewer#2

Comments from reviewer #2

  1. The manuscript would benefit from some additional editing to correct typographical errors. For example, p9 line 319 “... certain extend [extent]”, and several long sentences that could be edited for clarity. Generally, the manuscript reads clearly.

Authors sincerely appreciate reviewers for careful reading and constructive comments that improved this manuscript. We carefully checked and corrected typos. This revised paper was submitted after the commercial English editing service.

  1. There is a great deal of information dealing with Figure 1 and associated text in Section 2. Section 2 becomes rather dense on page 3 when describing human and budding yeast similarities. Might the authors consider a table to summarize the functions of subunits between yeasts and metazoans?  It may help clarify overall statements and alignment of roles/similarities/differences described.

Authors understand the reviewer’s point, but this manuscript is prepared for the special issue for yeast models of gene regulation. Therefore, a focus on human Spt16 should be kept to a minimum as it blurs the focus and changes the purpose of the special issue for yeast gene regulation.

  1. Figure 2- might the authors consider adding labels for subunits of FACT components/domains or an associated legend to reinforce connections of subunits with nucleosomes. This will help a broader readership to understand the structures described, particularly for HMGB-Nhp6.

As pointed out by reviewer#2, Figure 2 was modified.

  1. Figure 6 (and dinucleosome of figure 2)- linker DNA is mentioned; the authors might include indication of linker DNA in the diagram, along with labeled subunits.

As pointed out by reviewer#2, Figure 6 was modified.

  1. Additional summary of the applications of the work into humans/multicellular organisms may be considered in the summary and perspective (Section 8). Since FACT has both heterochromatic and transcriptional-activating roles (and across 2 yeast models), a few additional sentences to link these to human work (particularly given the differences from yeast to human, but also impact for human biomedical applications) could be particularly valuable.

We understand the point, but as we answered in the point No. 2, this paper was written for SI: yeast models for gene regulation, and it is not the main topic regarding the function of mammal FACT. In addition, according to the guideline for authors, the limitation is 4000 words for review manuscript, but this manuscript has already reached 4400 characters at this point. We think it is not appropriate to add more sentences.
